# Associations between metal constituents of ambient particulate matter and mortality in England: an ecological study

Aurore Lavigne,[1] Anna Freni Sterrantino [ORCID] ,[2] Silvia Liverani,[3] Marta Blangiardo,[2] Kees de Hoogh,[4,5] John Molitor,[6] Anna Hansell [ORCID] [7]

AL and AFS are joint first authors.

For numbered affiliations see end of article.

**Correspondence to**
Dr Anna Hansell;
ah618@leicester.ac.uk

## ABSTRACT

**Objectives** To investigate long-term associations between metal components of particulate matter (PM) and mortality and lung cancer incidence.

**Design** Small area (ecological) study.

**Setting** Population living in all wards (~9000 individuals per ward) in the London and Oxford area of England, comprising 13.6 million individuals.

**Exposure and outcome measures** We used land use regression models originally used in the Transport related Air Pollution and Health Impacts—Integrated Methodologies for Assessing Particulate Matter study to estimate exposure to copper, iron and zinc in ambient air PM. We examined associations of metal exposure with Office for National Statistics mortality data from cardiovascular disease (CVD) and respiratory causes and with lung cancer incidence during 2008–2011.

**Results** There were 108 478 CVD deaths, 48 483 respiratory deaths and 24 849 incident cases of lung cancer in the study period and area. Using Poisson regression models adjusted for area-level deprivation, tobacco sales and ethnicity, we found associations between cardiovascular mortality and $PM_{2.5}$ copper with interdecile range (IDR 2.6–5.7 ng/m$^3$) and IDR relative risk (RR) 1.005 (95%CI 1.001 to 1.009) and between respiratory mortality and $PM_{10}$ zinc (IDR 1135–153 ng/m$^3$) and IDR RR 1.136 (95%CI 1.010 to 1.277). We did not find relevant associations for lung cancer incidence. Metal elements were highly correlated.

**Conclusion** Our analysis showed small but not fully consistent adverse associations between mortality and particulate metal exposures likely derived from non-tailpipe road traffic emissions (brake and tyre wear), which have previously been associated with increases in inflammatory markers in the blood.

## INTRODUCTION

Chronic exposure to toxic substances in particulate matter (PM) with aerodynamic diameter less than 10 µm ($PM_{10}$)[1–3] and 2.5 µm ($PM_{2.5}$)[4] is associated with increased mortality levels from cardiovascular disease (CVD).[1 5] Some studies also show links between this long-term exposure to traffic-related air pollution and

## Strengths and limitations of this study

► One of the largest studies to explore exposure to metal components of ambient air in relation to mortality and lung cancer incidence, with 13.6 million population.

► A large number of cases: 108 478 cardiovascular disease deaths, 48 483 respiratory deaths and 24 849 incident cases of lung cancer in the study period and area, providing good statistical power to examine small excess risks.

► Established exposure models, developed and evaluated with measurements from a standardised monitoring campaign.

► An ecological study using registry data, without access to individual-level confounders other than age and sex.

► Metals were very highly correlated so multipollutant models could not be used.

lung cancer or respiratory mortality.[6] It has been suggested that metal components of PM may in part be responsible for toxic effects of air pollution on the cardiovascular and respiratory system.[7]

In the Transport related Air Pollution and Health Impacts—Integrated Methodologies for Assessing Particulate Matter (TRANSPHORM) study, copper zinc and iron content of PM ($PM_{10}$ and $PM_{2.5}$) were found to be associated—positively and significantly—with increases in inflammatory markers in the blood,[8] which might be expected to be associated with increased risks of cardiovascular and other diseases. However, a separate TRANSPHORM study[9] analysis of 19 cohorts with 9545 CVD deaths did not find any statistically significant associations with metal (or other) particulate components ($PM_{10}$ or $PM_{2.5}$). Here we use the same data sets to examine associations with mortality using a much larger data set than TRANSPHORM study,[9] to estimate particulate metal exposures for a population

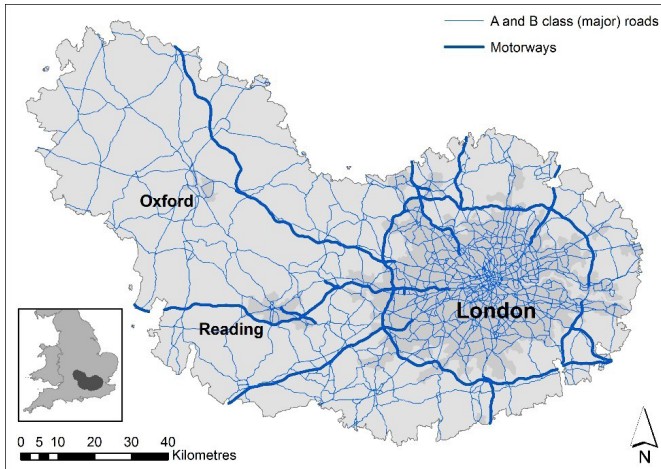

**Figure 1** Study area compromising London and Oxford areas, with major roads and motorways. In the inset the area localisation with regard to England map contains National Statistics data, Crown copyright and database right 2018; contains OS data, Crown copyright and database right 2018. All rights reserved.

of 13.6 million living in and near London, England, with 108478 CVD deaths and additionally 48 483 respiratory deaths and 24 849 incident cases of lung cancer.

## METHODS
Our study region covered a 10 782 km$^2$ area around London and Oxford (figure 1) in 1533 wards, an English Census area classification (primary unit of the English electoral geography) with a mean surface area ~7.0 km$^2$ and average 8892 inhabitants per ward, in our study period.

### Exposure data
In the region of London and Oxford PM was monitored during the years 2010–2011 as part of the European Study of Cohorts and Air Pollution Effects (ESCAPE) project.[10 11] Filters measuring PM$_{10}$ and PM$_{2.5}$ from the ESCAPE project were analysed for elemental composition and de Hoogh *et al* (2013)[12] developed land use regression (LUR) models for a number of the elemental components including metals as part of the TRANSPHORM project. These models were used to predict PM$_{10}$ and PM$_{2.5}$ elemental composition for our study population for 2010–2011. In brief, 20 sites were monitored for three 2-week periods[10] and PM$_{2.5}$ and PM$_{10}$ were separately collected using Harvard impactors. Their elemental composition was analysed using energy dispersive X-ray fluorescence. The association of PM elemental components with land use covariates relative to traffic, population, industry, or nature was evaluated with LUR models. Then, local estimates at the postcode level were predicted and aggregated at the super output area (SOA) level, with a population-weighted mean, for all SOAs in the study. Exposure was assigned for each case or incidence at postcode level.

In the analyses, we used copper (Cu), iron (Fe) and zinc (Zn) in the PM$_{10}$ fraction and copper and iron in the PM$_{2.5}$ fraction, all linked to non-tailpipe emissions.[12] LUR models for this selection of elements showed a good leave-one-out validation, explaining more than 77% (R$^2$) of the observed variability.

### Confounder data—deprivation, ethnicity and smoking data
To adjust for possible confounders in this study, we included area-level ethnicity from Census 2011 and accounted for percent of White and Asian people per ward as covariates in the models. We also used the 2007 index of multiple deprivation (IMD) as a relative measure of area-level deprivation (publicly available from the Department for Communities and Local Government data.gov.uk). This combines seven domains: 'income', 'employment', 'education', 'barriers to housing and services', 'crime', 'health' and 'living environment'. The latter is divided into two subdomains: 'indoor' measuring the quality of housing and 'outdoor' linked to air quality and road traffic accidents.[13] We excluded from the study the 'health' and 'outdoor living environment' domains,[14] since we examined associations between health outcomes and air pollution measures. The remaining domains were linearly combined to generate a 'modified IMD' relative score used in the analysis. High values of the modified IMD indicate higher deprivation. As a proxy for smoking, we used ward level tobacco expenditure (pounds/week/inhabitant) data obtained from CACI (CACI tobacco expenditure data are copyright 1996–2014 CACI Ltd.).

### Health data
Mortality counts for cardiovascular (International Classification of Diseases, Tenth Revision (ICD10) I00-I99) and respiratory (CDC10 J00-J99) disease and lung cancer incidence counts (C33 and C34 ICD10 codes) were extracted for 2008–2011 from Office National Statistics data held by the Small Area Health Statistics Unit, which provide 100% coverage of deaths. The counts were then adjusted by sex and 5-year age band.

### Patient and public involvement
Patients were not involved in the development of the research question or the design and conducting of the study.

### Statistical analysis
The effect of PM exposure to copper, iron and zinc on health outcomes was analysed with Poisson regression (a generalised linear model) of count data at small area (ward) level, implemented in a Bayesian framework with spatial residuals, see online supplementary figure 1 for a graphical representation of the possible causal mechanism.

Let $Y_i$ denote the number of cases recorded in the spatial unit $i$ and $E_i$ the expected count taking into account the age and sex structure of the population at risk (internal standardisation). Then, using Poisson regression, $Y_i$ is

assumed to follow a Poisson distribution with mean equal to $E_iRR_i$ such that

$$\log(RR_i) = \mu + \sum_{j=1}^{p_1} \alpha_j Confound_{ij} + \beta PM_{ik} + U_i$$

Here, $\mu$ is the model intercept, $Confound_{ij}$ denotes the value of the confounder $j$ $(1, \ldots, p_1)$ for area $i$ $(1, \ldots, n)$; similarly $PM_{ik}$ stands for the PM $k$ $(1, \ldots, p_2)$ exposures, $U_i$ is a spatial random effect, modelled with an intrinsic conditional autoregressive model,[15] accounting for the spatial dependence of residuals. The coefficients $\alpha_j$ and $\beta$ indicate the linear effect of the confounders and PM metals on the log relative risk (RR).

For each health outcome, the analysis was performed separately for elemental constituents of $PM_{10}$ and $PM_{2.5}$. A second model was fitted, for each PM metal constituents and as measure of multicollinearity and variation inflation factor (VIF) is provided.

Both models are inferred using the Bayesian approach in R-package Integrated Nested Laplace Approximations (INLA).[16] We used the non-informative priors proposed as default in R-INLA and standardised confounders.

Regression parameters are expressed per interdecile range (IDR) RR, that is, the increase of the RR when the level of covariates increases from the 10th to 90th centile; the posterior mean and 95% credible bounds are given.

## RESULTS

There were 108 478 cardiovascular deaths, 48 483 respiratory deaths and 24 849 incident lung cancer cases in the study area for 2008–2011 (table 1 and online supplementary table S1). We have reported summary descriptive statistics for standard mortality/incidence rates (SMR/SIRs), metal constituents of PM and confounders, stratifying the wards between the 10th percentile of exposure 90th percentile of $PM_{2.5}$ copper. SMRs/SIRs, metal constituents of PM, area-level deprivation, non-white ethnicity and tobacco sales (smoking proxy) were all higher in wards in the 90th versus 10th percentile $PM_{2.5}$ copper.

Maps of the spatial distribution of the covariates and elemental concentrations show that highest values were in greater London area, with iron and zinc also high in wards with motorways (figure 2). The percentage population ethnicity for wards had a median of 77% white and 9% Asian ethnicity (predominantly of South Asian origin). Most of the areas with low percentage of white population were concentrated in greater London, which also had higher percentage of Asian (online supplementary figure S2).

The individual linear effect of each elemental constitute of PM evaluated with the Poisson regression adjusted for confounders is displayed in table 2 and online supplementary table S2 in supplementary material. Statistically significant associations with PM metal concentrations were identified for cardiovascular and respiratory mortality but not lung cancer incidence. For cardiovascular mortality, copper in the $PM_{2.5}$ fraction was associated with a small

increased RR 1.005 (95%CI 1.001 to 1.009) per IDR but iron had an apparent protective association (RR 0.042 95% CI 0.002 to 0.995) although with extremely high uncertainty. For respiratory mortality, the copper in the $PM_{10}$ fraction had a very small protective association (RR 0.988 95% CI 0.978, 0.998), but $PM_{10}$ zinc was associated with an increased mortality risk (RR 1.136 95% CI 1.010 to1.277).

The elements were highly correlated: 0.88 for $PM_{2.5}$ elements and 0.82–0.92 for $PM_{10}$ elements (table 3). For $PM_{10}$ the Pearson correlation between copper and zinc was 0.85, and for $PM_{2.5}$ the correlation between copper and iron was 0.88. The metal constituents showed high correlation with $PM_{2.5}$ and $PM_{10}$ mass concentrations for $PM_{2.5}$ and metals in $PM_{2.5}$ were 0.86–0.89 and 0.73–0.89 for $PM_{10}$ metals; for $PM_{10}$ and $PM_{10}$ metals 0.74–0.88 and 0.86–0.89 for metals in $PM_{2.5}$ (see online supplementary table S3).

Thus, it is not possible to definitively attribute an association with one metal element given the interdependence.

In the model fit, for each group of metals by PM, we have found that area-level deprivation (IMD) and weekly tobacco spend had a clear adverse association with cardiovascular mortality, respiratory mortality and lung cancer incidence (online supplementary table S3), with moderate high value of VIF. On the contrary, the proportions of White and Asian people in wards were associated with lower risks for the three diseases, suggesting a weak influence of the ethnic composition of the population on mortality/incidence rate.

## DISCUSSION

This ecological study at small area level examined associations between modelled particulate metal (copper, iron and zinc) concentrations in relation to cardiovascular and respiratory mortality and lung cancer incidence in and around Greater London covering 13.6 million population with approximately 110 000 cardiorespiratory deaths and 25 000 new lung cancer cases. While the results did not find evidence of positive association between ambient particulate metal concentrations and lung cancer incidence, Poisson regression suggested copper in the $PM_{2.5}$ fraction had statistically significant association with increased cardiovascular mortality risk and $PM_{10}$ zinc with respiratory mortality risk. Results for metal constituents were not fully consistent within our study for the same element in $PM_{2.5}$ and $PM_{10}$ size fractions. Metal exposures were highly correlated, so it is difficult to definitively attribute an association with one metal element.

Advantages of our study include the use of extremely large data sets with population coverage giving good statistical power to detect even very small associations. Another advantage was the use of standardised exposure models developed from standardised monitoring campaigns to estimate spatial variability in long-term exposures. While exposure data were derived from LUR models that showed good predictability, they may

**Table 1** Descriptive statistics of health outcomes, modelled particulate metal concentrations, deprivation score and ethnicity covariates for the 1533 wards in the study area in 2008–2011, subdivided by PM$_{2.5}$ copper <10th, 10th–90th and >90th quantile

| | Cu* PM$_{2.5}$ | | | | | | | | | | | |
|---|---|---|---|---|---|---|---|---|---|---|---|---|
| | 10th centile (n=153) | | | | 10th–90th centile (n=1225) | | | | >90th centile (n=154) | | | |
| | 10th centile | Mean | Median | 90th centile | 10th centile | Mean | Median | 90th centile | 10th centile | Mean | Median | 90th centile |
| **Health outcomes** | Standard mortality/incidence ratio (ratio across whole study area=1.00) | | | | | | | | | | | |
| Cardiovascular mortality | 0.57 | 0.83 | 0.86 | 1.13 | 0.72 | 0.99 | 1.01 | 1.33 | 0.63 | 0.96 | 1.00 | 1.35 |
| Respiratory mortality | 0.46 | 0.81 | 0.81 | 1.20 | 0.61 | 0.98 | 1.02 | 1.46 | 0.50 | 0.94 | 0.94 | 1.36 |
| Lung cancer incidence | 0.40 | 0.81 | 0.86 | 1.33 | 0.53 | 0.95 | 0.99 | 1.53 | 0.64 | 1.14 | 1.16 | 1.73 |
| **Area-level confounders** | | | | | | | | | | | | |
| Modified IMD | 3.26 | 4.85 | 5.15 | 7.41 | 3.35 | 6.29 | 6.83 | 11.30 | 7.91 | 11.22 | 10.99 | 13.24 |
| % of Asian | 0.01 | 0.01 | 0.01 | 0.02 | 0.02 | 0.07 | 0.12 | 0.26 | 0.07 | 0.11 | 0.15 | 0.31 |
| % of White | 0.95 | 0.97 | 0.97 | 0.99 | 0.46 | 0.86 | 0.78 | 0.95 | 0.43 | 0.62 | 0.60 | 0.77 |
| Tobacco expenditure (pounds/week/inhabitant) | 3.19 | 3.88 | 3.96 | 4.86 | 3.41 | 4.46 | 4.57 | 5.96 | 4.45 | 5.65 | 5.53 | 6.55 |

*Cu PM$_{2.5}$ metals in ng/m$^3$; LOOCV R$^2$=0.79.
†LOOCV.
‡LUR.
IMD, index of multiple deprivation; LOOCV, leave-one-out cross-validation; LUR, land use regression; PM, particulate matter.

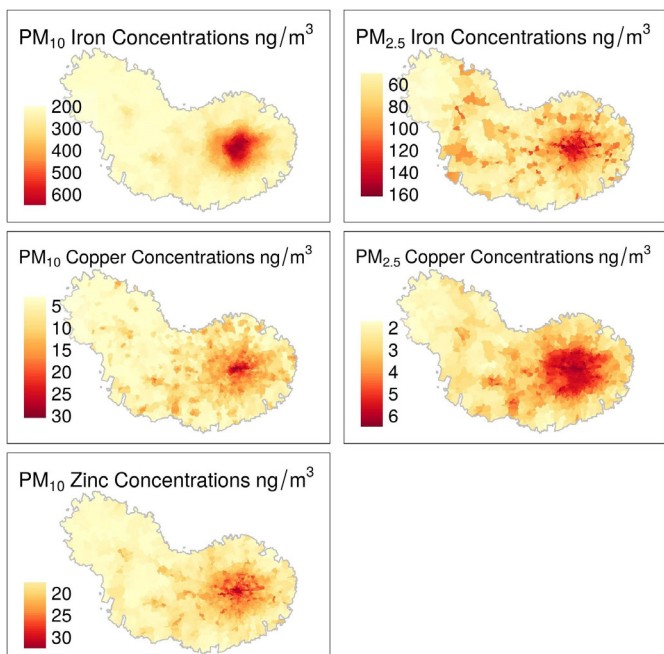

**Figure 2** Maps of the metal exposures population weighted by ward. Contains National Statistics data, Crown copyright and database right 2018; contains OS data, Crown copyright and database right 2018. All rights reserved.

still misclassify true exposure as (i) prediction is good but not perfect (ii) using a model of exposure at residence as a proxy for personal exposure. A limitation

**Table 2** Individual effects of metals, estimated with Poisson regression, on cardiovascular mortality, respiratory mortality and lung cancer incidence adjusted for tobacco weekly expenditure, IMD and percentage of Asian and white population. Mean and lower and upper bounds of the credible intervals of the IDR RR

|  | Metal | RR (95% CI) |
|---|---|---|
| Cardiovascular mortality | Cu PM$_{10}$ | 0.994 (0.987 to 1.001) |
|  | Fe PM$_{10}$ | 0.319 (0.037 to 2.779) |
|  | Zn PM$_{10}$ | 1.073 (0.985 to 1.169) |
|  | Cu PM$_{2.5}$ | 1.005 (1.001 to 1.009) |
|  | Fe PM$_{2.5}$ | 0.042 (0.002 to 0.995) |
| Respiratory mortality | Cu PM$_{10}$ | 0.988 (0.978 to 0.998) |
|  | Fe PM$_{10}$ | 0.649 (0.033 to 12.767) |
|  | Zn PM$_{10}$ | 1.136 (1.010 to 1.277) |
|  | Cu PM$_{2.5}$ | 1.003 (0.998 to 1.009) |
|  | Fe PM$_{2.5}$ | 0.980 (0.013 to 72.673) |
| Lung cancer incidence | Cu PM$_{10}$ | 0.998 (0.912 to 1.091) |
|  | Fe PM$_{10}$ | 0.973 (0.830 to 1.142) |
|  | Zn PM$_{10}$ | 0.995 (0.910 to 1.089) |
|  | Cu PM$_{2.5}$ | 1.092 (0.943 to 1.225) |
|  | Fe PM$_{2.5}$ | 0.969 (0.889 to 1.057) |

IDR, interdecile range; IMD, index of multiple deprivation; PM, particulate matter; RR, relative risk.

in our exposure assessment is the limited number of monitoring sites, 20, which potentially can lead to overfitting of the developed LUR models.[17] Providing that densities of measurement sites and estimation sites (wards) are similar, Szpiro and Paciorek (2013)[18] show that in case of over-smoothing of the exposure, the association between outcomes and exposure may be underestimated. In our case, over-smoothing likely occurs and this issue may partially explain our difficulty to show evidence of adverse associations between health outcomes and exposures to particulate elements. As most other ambient air pollution studies, we use outdoor concentration of pollutants at residence, without taking into account indoor levels, travel exposure or places of work. The correlation between indoor and outdoor concentration is high for fine particulate (PM$_{2.5}$),[19] suggesting that ignoring the indoor concentration is a small issue. However, in the London region, the difference of exposure at home and workplace may be different, since a part of the population living in suburban areas work in the city centre, where exposures are higher. Another limitation is that we used LUR models predicting particulate metals in 2010–2011 to look at associations with mortality during 2008–2011. Our exposure estimates should also be representative of the preceding 2 years and should capture deaths related to short/intermediate/long-term influences. However, we used an ecological study design with limited ability to control for confounders at the individual level.

There are a limited number of other health studies looking at copper, zinc and iron metal components of particulates. Three studies looking at long-term effects using similarly derived estimates from the TRANS-PHORM project as used here but much smaller numbers of health events than this study found significant associations with inflammatory markers in blood but not health events. Hampel *et al*[8] found positive statistically significant associations between PM$_{2.5}$ copper and PM$_{10}$ iron with high-sensitivity C-reactive protein and PM$_{2.5}$ zinc with fibrinogen in five European cohorts with available biomarkers (>17 000 measurements). Wolf *et al*[20] found elevated but non-significant positive associations with copper, zinc and iron constituents of particulates (PM$_{10}$ or PM$_{2.5}$) with incident coronary events in 11 cohorts (5157 events), while Wang *et al*[9] did not find long-term positive associations with cardiovascular mortality (9545 deaths) in 19 European cohorts where exposure results from a single year were applied over 2–20 years follow-up, in some cases retrospectively. A further study, the California teachers study[21] found positive and significant associations between PM$_{2.5}$ copper estimated in 2001–2007 and contemporaneous ischaemic heart disease deaths (1085 events) and elevated but non-significant associations with PM$_{2.5}$ iron and other metals.

We did not find associations with lung cancer incidence. While toxicological studies suggest that metals in airborne particulates are genotoxic,[22] the reason we did not find an association even in our large sample

**Table 3** Pearson intercorrelation (r) between PM metals (n=1533)

|  | PM$_{10}$ copper | PM$_{10}$ iron | PM$_{10}$ zinc | PM$_{2.5}$ iron | PM$_{2.5}$ copper |
|---|---|---|---|---|---|
| PM$_{10}$ copper | 1 |  |  |  |  |
| PM$_{10}$ iron | 0.85 | 1 |  |  |  |
| PM$_{10}$ zinc | 0.85 | 0.92 | 1 |  |  |
| PM$_{2.5}$ iron | 0.82 | 0.91 | 0.93 | 1 |  |
| PM$_{2.5}$ copper | 0.75 | 0.89 | 0.90 | 0.88 | 1 |

PM, particulate matter.

size may be because our exposure measures relate to a similar time frame as the health outcome. Studies finding associations of particulates with lung cancer have typically considered 10 or more years follow-up.[23]

Short-term associations of metal components of particulates with mortality were examined in a systematic review of time series studies of fine-particle components and health published up to 2013.[24] Zinc, indicative of road dust and possibly a result of tyre wear, was associated with daily mortality in 8 of 11 studies included in the review. The subsequently published MED-PARTICLES time-series analysis in five European cities Basagaña *et al*[17] found positive significant short-term associations with PM$_{10}$ copper iron and zinc and PM$_{2.5}$ iron with cardiovascular hospitalisations and PM$_{10}$ and PM$_{2.5}$ zinc for respiratory disease hospitalisations, but no significant associations were seen for mortality.

The reason that results for metal constituents of particulates are not completely consistent across studies may be that metal concentrations serve as a proxy for oxidative potential.[25] Within the study area and in the analysis, the TRANSPHORM metal particulate measurements used to derive the LUR models were highly correlated with oxidative potential of the particulates as measured using ascorbate (Pearson's r=0.93 for copper, 0.95 for iron, 0.67 for zinc).[25] The high correlations between metal constituents of particulates raise the possibility that observed associations for one metal actually relate to another element that was better estimated. The high correlations also preclude conducting multipollutant analyses using Poisson regression.

## CONCLUSION
We found positive and significant associations suggestive of small increased risk of cardiovascular and respiratory mortality but not lung cancer incidence in Greater London and surroundings in relation to metal concentrations of ambient particulate matter, which are likely derived from non-tailpipe road traffic emissions (brake and tyre wear). Findings are consistent with a previous study finding associations of particulate metals with inflammatory markers, but further work is needed to better define exposures to airborne metal elements and non-tailpipe emissions.

**Author affiliations**
[1]UFR MIME, Domaine universitaire du Pont de Bois, Université de Lille 3 UFR MIME, Villeneuve-d'Ascq, Nord-Pas-de-Calais-Picard, France
[2]Department of Epidemiology and Biostatistics, MRC-PHE Centre for Environment and Health, School of Public Health, Imperial College London, London, UK
[3]School of Mathematical Sciences, Queen Mary University of London, London, UK
[4]Swiss Tropical and Public Health Institute, Basel, Switzerland
[5]University of Basel, Basel, Switzerland
[6]School of Biological and Population Health Sciences, College of Public Health and Human Sciences, Oregon State University CAPS, Corvallis, Oregon, USA
[7]University of Leicester, Leicester, UK

**Acknowledgements** We would like to acknowledge and thank Professor John Gulliver, Dr Gary Fuller, Dr David Morley and Professor Nicky Best for their useful comments. CACI tobacco expenditure data are © Copyright 1996-2014 CACI Ltd.

**Contributors** AL and AFS drafted the paper and ran the statistical analyses. KdH provided exposure data. SL, JM and MB advised on the statistical methods. AH designed the study. All the authors provided intellectual input, interpreted the results and helped to revise the manuscript. All authors approved the final version of the manuscript and agreed to be accountable for all the aspects of the work in ensuring that questions related to the accuracy or integrity of any part of the work are appropriately investigated and resolved. AH is the guarantor of this paper.

**Funding** The research project was funded through Medical Research Council (MRC) (grant G09018401) and the Small Area Health Statistics Unit (SAHSU). The work of the UK SAHSU is funded by Public Health England as part of the MRC-Public Health England (PHE) Centre for Environment and Health, funded also by the UK Medical Research Council. The air pollution exposure assessments used in the research leading to these results were funded by the European Community's Seventh Framework Program (FP7/2007-2011), European Study of Cohorts and Air Pollution Effects projects (grant agreement 211250) and Transport related Air Pollution and Health Impacts—Integrated Methodologies for Assessing Particulate Matter. The research was funded/part funded by the National Institute for Health Research Health Protection Research Unit in Health Impact of Environmental Hazards at King's College London in partnership with PHE and Imperial College London.

**Competing interests** None declared.

**Patient consent for publication** Not required.

**Ethics approval** Small Area Health Statistics Unit holds approvals from the National Research Ethics Service—reference 12/LO/0566 and 12/LO/0567—and from the Health Research Authority Confidentially Advisory Group (HRA-CAG) for Section 251 support (HRA-14/CAG/1039) for use of the health data used in this research.

**Provenance and peer review** Not commissioned; externally peer reviewed.

**Data availability statement** Data may be obtained from a third party and are not publicly available.

**ORCID iDs**
Anna Freni Sterrantino http://orcid.org/0000-0002-6602-6209
Anna Hansell http://orcid.org/0000-0001-9904-7447

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
