## [Reviewer comments · BMJ Open]

ARTICLE DETAILS

TITLE (PROVISIONAL)	Associations between metal constituents of ambient particulate matter and mortality in England; an ecological study
AUTHORS	Lavigne, Aurore; Freni Sterrantino , Anna ; Liverani, Silvia; Blangiardo, Marta; de Hoogh, Kees; Molitor, John; Hansell, Anna

VERSION 1 – REVIEW

REVIEWER	R. Bruce Urch, Adjunct Professor Division of Occupational & Environmental Health, Dalla Lana School of Public Health, University of Toronto, Toronto, Ontario, Canada
REVIEW RETURNED	04-Apr-2019

GENERAL COMMENTS	Overview. This manuscript describes an ecological study investigating associations between metal constituents of PM10 (Cu, Zn & Fe) as well as PM2.5 (Cu & Fe) and CVD & respiratory mortality and lung cancer incidence in the London and Oxford area of England. The study region covered a 10,782 km² area comprising 13.6 million individuals. Over the study period (2008-2011), there were 108,478 CVD and 48,483 respiratory deaths and 24,849 incident lung cancer cases. LUR models originally used in the TRANSPHORM study were used to estimate exposure to Cu, Zn & Fe in ambient PM. Health outcome data were obtained from the Office for National Statistics. Results for Poisson regression models showed a positive association between CVD mortality and PM2.5 Cu levels as interdecile range relative risk (RR=1.005, 95% CL=1.001, 1.009) and between respiratory mortality and PM10 Zn levels (RR=1.136, 95% CL: 1.010, 1.277). Models were adjusted for area-level deprivation, tobacco sales and ethnicity. No relevant associations were seen for lung cancer incidence. The authors concluded that there were small but not fully consistent adverse associations between metal exposure, likely from brake/tire wear, and mortality. The authors also concluded that the findings were consistent with a previous study finding associations of metals with inflammatory markers. General Comments: The paper is well written and an important addition to the literature. There are some minor points of clarification and suggestions detailed in the specific comments. Throughout manuscript. Be consistent with use of subscripts for PM10 and PM2.5. When you mention associations, specify if they are positive or negative and if they are significant.
---

	Specific Comments: Results are presented for three elements: Cu, Fe and Zn analyzed by XRF. Please expand on why only these three were selected. Where other elements not measured (Tsai et al paper reports 17 “well detected” elements measured by XRF in PM2.5 & PM10 in ESCAPE study), too low or no LUR models? Introduction. Pg 3, lines 10-12. “In the Transport related Air Pollution and Health impacts – Integrated Methodologies for Assessing Particulate Matter (TRANSPHORM) study, copper zinc and iron content of particulate matter ...”. Specify what size fraction of PM. The same for the Wang et al study (ref 9). Pg 3, lines 16/17. “Here we use the same datasets examine associations ...”. Missing “to”, “datasets to examine”. Pg 3, lines 16/17. Same sentence as above “... using a much larger dataset ...”. What was the size of the dataset used by Wang et al? Methods. Pg 3, lines 27-30. Specify that the mean surface area and average inhabitants was per ward. Exposure data. Page 3, line 48. Give a reference for “... all linked to non-tailpipe emissions.” Statistical analysis. Pg 4. Please specify the test used for correlations between metals. Pearson? Results. Table 1. Pg 5. Some formatting issues. e.g. median- “n” on a 2nd line. Closing bracket “)” for pounds/week/inhabitant on a 2nd line. PM10 has subscript, PM2.5 does not. Define the acronyms used in a footnote below the Table. e.g., LOOCV, LUR. Table 2. Pg 6. Spell out IMD. PM10 has subscript, PM2.5 does not. Results. Pg 7, line 6. Specify that it is “r” you are reporting as opposed to “r2”. As per pg 8, line 50. “(Pearson r = 0.93 for copper ...” Pg 7, line 6. “The elements were highly correlated: 0.88 for PM2.5 elements and 0.88-0.92 for PM10 elements (Table 3).” Should be 0.85-0.92 for PM10. Pg 7, lines 25/26. “Area-level deprivation ...”. Modify to “Area-level deprivation (IMD) ...”, as Table S1 uses IMD. Pg 7, lines 27-30. “On the contrary, the proportions of White and Asian people in wards was associated with lower risks for the three diseases, suggesting a weak influence of the ethnic composition of the population on mortality/incidence rate”. What
--	--

	were the other ethnic groups? Where the numbers large enough to examine associations to see if risks were higher for them? Discussion. You do not discuss the lack of significant findings for lung cancer incidence. Please provide some hypothesis as to why not. e.g., study period too short. You had rational to include it as an outcome. Was this a primary, secondary or tertiary outcome? Pg 7, lines 43-45. "However, results for metal constituents were not fully consistent within our study." Explain what you mean by "not fully consistent". Three possibilities: Some adverse and some beneficial effects; PM10 vs PM2.5; or Cu vs Fe vs Zn. Pg 7, line 52. Explain/expand what you mean by "misclassify true exposure". Pg 7, lines 57-58. "In our case, over-smoothing likely occurs and this issue may partially explain our difficulty to show evidence of associations between health outcomes and exposures to particulate elements." Do you mean "significant adverse associations" as you also found some protective associations? Please clarify what you mean by "evidence of associations". Pg 8. When you mention associations, specify if they are positive or negative and if they are significant.
--	--

REVIEWER	Weichenthal, Scott McGill University Department of Epidemiology Biostatistics and Occupational Health
REVIEW RETURNED	26-Apr-2019

GENERAL COMMENTS	The authors conducted an ecological study of mortality/lung cancer incidence and metal concentrations in particulate air pollution in England. Outdoor PM-metal concentrations were estimated using an LUR model developed from a small number of sites (20) in the study area in 2010-2011. All of the exposures are very highly correlated, making it difficult to identify specific metals of interest. The ecological nature of the study and very high correlations between PM-metals limits the significance of this contribution. Specific comments  1. The abstract should indicate the concentration interval corresponding the health risk estimates. 2. The authors should provide a DAG illustrating how the conceptual framework linking the exposure of interest (outdoor PM-metal concentrations) to mortality/cancer including confounding factors. Some of the parameters included as confounders are likely not causes of outdoor PM-metals (i.e. tobacco sales). A DAG would help to clarify the thinking behind the selection of covariates. 3. How large were the spatial units of analysis? Were they of similar size across the study area? 4. The authors should show RRs for PM2.5 and correlations between the PM-metals and PM mass concentrations.
--

REVIEWER	Justin Manjourides
-----------------	--------------------

	Northeastern University
REVIEW RETURNED	23-May-2019

GENERAL COMMENTS	Associations between metal constituents of ambient particulate matter and mortality in England; a small area study The authors aim to assess long-term association between components of PM10 and PM2.5 and mortality and lung cancer incidence. Land use regression models estimate exposures, while health outcomes are derived from the Office of National Statistics. Overall, I found this paper of interest and with sound methods. I do think this work could benefit from more clarity around the following topics:  1. A slightly more expanded explanation of the exposure assignment. I'm gathering from Page 3 line 44 that participants were matched to the exposure estimate for their postal code. This could be made more explicit. 2. Your definition of "long term exposure" was never clearly stated. 3. I'm not an exposure scientist, so I am curious as to why Zinc was included in the PM10 analysis, but not the PM25 analysis. 4. The model on Page 4 indicates that all PM10 constituents were included in the same model and all Pm25 constituents were included in the same model. However, the correlations between these constituents, within each PM10 and PM25, are extremely high. While this is certainly a recognized issue in air pollution research, it would be good to see perhaps single-constituent models as a comparison or sensitivity analysis. 5. The μ in the model is not defined. 6. Be more explicit that the main results presented control for the listed confounders, also detailing these control variables in the table caption. 7. Throughout this manuscript I would suggest emphasizing that these are all estimates of the linear association between the log(RR) and AP exposures, as there could very well be a different non-linear association caused by threshold effects or effect modification. 8. Summary data in Table 1 display 10th and 90th percentile for each metal and confounder. While this is informative for the metals, to get a sense of the exposures, this is less informative for the confounders. The important comparison is what the values of the outcomes and confounders are for those individuals living in areas that have the 10th percentile of exposure vs those that individuals living in the areas that have the 90th percentile of exposure. This will help tell the story of how those high exposure areas differ from those low exposure areas on key demographic features and outcome rates. I would suggest a slight reorganization of Table 1 where the columns represent the 10th and 90th percentiles of either PM10 and PM25 or, because the correlations are so high, perhaps selecting 1 constituent of PM10 and one of PM25 as an exemplar, and stratify your sample demographics and outcomes on that variable. Minor:  1. Page 4, Line 35: "... for elemental constituents of in PM10 and PM25."
---

VERSION 1 – AUTHOR RESPONSE

Reviewer: 1

Reviewer Name: R. Bruce Urch, Adjunct Professor

Institution and Country: Division of Occupational & Environmental Health, Dalla Lana School of Public Health, University of Toronto, Toronto, Ontario, Canada

Please state any competing interests or state 'None declared': None declared

Please leave your comments for the authors below

Overview.

This manuscript describes an ecological study investigating associations between metal constituents of PM₁₀ (Cu, Zn & Fe) as well as PM_{2.5} (Cu & Fe) and CVD & respiratory mortality and lung cancer incidence in the London and Oxford area of England. The study region covered a 10,782 km² area comprising 13.6 million individuals. Over the study period (2008-2011), there were 108,478 CVD and 48,483 respiratory deaths and 24,849 incident lung cancer cases. LUR models originally used in the TRANSPHORM study were used to estimate exposure to Cu, Zn & Fe in ambient PM. Health outcome data were obtained from the Office for National Statistics. Results for Poisson regression models showed a positive association between CVD mortality and PM_{2.5} Cu levels as interdecile range relative risk (RR=1.005, 95% CL=1.001, 1.009) and between respiratory mortality and PM₁₀ Zn levels (RR=1.136, 95% CL: 1.010, 1.277). Models were adjusted for area-level deprivation, tobacco sales and ethnicity. No relevant associations were seen for lung cancer incidence. The authors concluded that there were small but not fully consistent adverse associations between metal exposure, likely from brake/tire wear, and mortality. The authors also concluded that the findings were consistent with a previous study finding associations of metals with inflammatory markers.

General Comments:

The paper is well written and an important addition to the literature. There are some minor points of clarification and suggestions detailed in the specific comments.

We would like to thank Prof. Bruce Urch for the comments, and we have addressed the minor points and replied to comments as follows.

Throughout manuscript.

Be consistent with use of subscripts for PM₁₀ and PM_{2.5}.

When you mention associations, specify if they are positive or negative and if they are significant.

We had made the changes for the subscripts and added the positive or negative specification when association has been mentioned.

Specific Comments:

Results are presented for three elements: Cu, Fe and Zn analyzed by XRF. Please expand on why only these three were selected. Where other elements not measured (Tsai et al paper reports 17 "well detected" elements measured by XRF in PM_{2.5} & PM₁₀ in ESCAPE study), too low or no LUR models?

From the larger list of elements detected by XRF we only developed LUR models for "Cu, Fe, and Zn mainly for (no tailpipe) traffic emissions; S for long-range transport; Ni and V for mixed oil burning/industry; Si for crustal material; and K for biomass burning" (de Hoogh et al. 2013). In the London and Oxford region models for Cu, Fe and Zn were robust, whereas models for S, Ni, V, Si and K were poor predictors of concentrations so we did not include in the analysis.

Introduction.

Pg 3, lines 10-12. "In the Transport related Air Pollution and Health impacts – Integrated Methodologies for Assessing Particulate Matter (TRANSPHORM) study, copper zinc and iron content of particulate matter ...". Specify what size fraction of PM. The same for the Wang et al study (ref 9). We added this.

Pg 3, lines 16/17. "Here we use the same datasets examine associations ...". Missing "to", "datasets to examine".

Corrected.

Pg 3, lines 16/17. Same sentence as above "... using a much larger dataset ...". What was the size of the dataset used by Wang et al?

Yes, we made it clear.

Methods.

Pg 3, lines 27-30. Specify that the mean surface area and average inhabitants was per ward.

We added the range in the study area paragraph.

Exposure data.

Page 3, line 48. Give a reference for "... all linked to non-tailpipe emissions."

We added it.

Statistical analysis.

Pg 4. Please specify the test used for correlations between metals. Pearson?

Yes, we used Pearson and we add it in the text.

Results.

Table 1. Pg 5. Some formatting issues. e.g. median- "n" on a 2nd line. Closing bracket ")" for pounds/week/inhabitant on a 2nd line. PM10 has subscript, PM2.5 does not. Define the acronyms used in a footnote below the Table. e.g., LOOCV, LUR.

We specified the acronyms as footnote and amended the rest.

Table 2. Pg 6. Spell out IMD. PM10 has subscript, PM2.5 does not.

Corrected.

Results.

Pg 7, line 6. Specify that it is "r" you are reporting as opposed to "r²". As per pg 8, line 50.

"(Pearson r = 0.93 for copper ..."

Corrected.

Pg 7, line 6. "The elements were highly correlated: 0.88 for PM2.5 elements and 0.88-0.92 for PM10 elements (Table 3)." Should be 0.85-0.92 for PM10.

We adjusted it.

Pg 7, lines 25/26. "Area-level deprivation ...". Modify to "Area-level deprivation (IMD) ...", as Table S1 uses IMD.

Corrected.

Pg 7, lines 27-30. "On the contrary, the proportions of White and Asian people in wards was associated with lower risks for the three diseases, suggesting a weak influence of the ethnic composition of the population on mortality/incidence rate". What were the other ethnic groups? Where the numbers large enough to examine associations to see if risks were higher for them?

Central London has a very diverse ethnic population with white, Asian (here mainly South Asian from Indian subcontinent) and black African groups being predominant. However, the Oxford is predominantly white. There we have used in our analyses the main groups – white, Asian and other - across the study region as a whole.

Discussion.

You do not discuss the lack of significant findings for lung cancer incidence. Please provide some hypothesis as to why not. e.g., study period too short. You had rational to include it as an outcome. Was this a primary, secondary or tertiary outcome?

We have added the following to the discussion "We did not find associations with lung cancer incidence. While toxicological studies suggest that metals in airborne particulates are genotoxic(Bocchi et al. 2019), the reason we did not find an association even in our large sample size may be because our exposure measures relate to a similar time frame as the health outcome. Studies finding associations of particulates with lung cancer have typically considered 10 or more years follow-up (Raaschou-Nielsen et al. 2016)".

Pg 7, lines 43-45. "However, results for metal constituents were not fully consistent within our study." Explain what you mean by "not fully consistent". Three possibilities: Some adverse and some beneficial effects; PM10 vs PM2.5; or Cu vs Fe vs Zn.

Not fully consistent means for example, that while we saw an adverse effect for copper PM2.5 but not PM10. We added in the text: "Results for metal constituents were not fully consistent within our study for the same element in PM2.5 and PM10 size fractions. Metal exposures were highly correlated so it is difficult to definitively attribute an association with one metal element."

Pg 7, line 52. Explain/expand what you mean by "misclassify true exposure".

As (i) prediction is good but not perfect (ii) using a model of exposure at residence as a proxy for personal exposure. This has been added to the text.

Pg 7, lines 57-58. "In our case, over-smoothing likely occurs and this issue may partially explain our difficulty to show evidence of associations between health outcomes and exposures to particulate elements." Do you mean "significant adverse associations" as you also found some protective associations? Please clarify what you mean by "evidence of associations".

We were thinking of adverse associations and have clarified the text accordingly.

Pg 8. When you mention associations, specify if they are positive or negative and if they are significant.

We added it in the text if the associations were negative or positive.

Reviewer: 2

Reviewer Name: Scott Weichenthal

Institution and Country: McGill University Department of Epidemiology Biostatistics and Occupational Health

Please state any competing interests or state 'None declared': None declared

We would like to thank Prof. Weichenthal for the comments, which we have addressed below.

Please leave your comments for the authors below

The authors conducted an ecological study of mortality/lung cancer incidence and metal concentrations in particulate air pollution in England. Outdoor PM-metal concentrations were estimated using an LUR model developed from a small number of sites (20) in the study area in 2010-2011. All of the exposures are very highly correlated, making it difficult to identify specific metals of interest. The ecological nature of the study and very high correlations between PM-metals limits the significance of this contribution.

Specific comments

1. The abstract should indicate the concentration interval corresponding the health risk estimates. We added the interdecile values for the PM10 zinc and PM2.5 copper, in the abstract.

2. The authors should provide a DAG illustrating how the conceptual framework linking the exposure of interest (outdoor PM-metal concentrations) to mortality/cancer including confounding factors. Some of the parameters included as confounders are likely not causes of outdoor PM-metals (i.e. tobacco sales). A DAG would help to clarify the thinking behind the selection of covariates. We added a DAG as supplementary figure(S2), where we provided a possible causal mechanism for the confounders. We have hypothesized that conditioning on ethnicity this influence the deprivation and the smoking attitude and exposure. All have an have effect on the adverse health outcomes, with deprivation directly affecting exposure as well.

3. How large were the spatial units of analysis? Were they of similar size across the study area? The spatial units are the wards of size 0.12 - 104.3 Km², with mean and median respectively of 7.0 and 2.5 Km². The wards in terms of area dimension are different as they are defined within a local authority area and typically each contain roughly the same number of electors, and each elect three councillors.

4. The authors should show RRs for PM2.5 and correlations between the PM-metals and PM mass concentrations.

The table below shows the correlation between the adjusted annual mean concentrations of PM-metals (only the ones we used) and the adjusted annual mean PM concentrations (PM2.5 and PM10). We also added this table into the supplementary material.

	PM2.5 CU	PM2.5 FE	PM10 CU	PM10 FE	PM10			
ZN	PM2.5	PM10						
PM2.5	Correlation	.862**	.899**	.896**	.895**	.731**	1	.925**
	p-value	<0.001	<0.001	<0.001	<0.001		<0.001	
PM10	Correlation	.825**	.877**	.866**	.889**	.747**	.925**	1
	p-value	<0.001	<0.001	<0.001	<0.001	<0.001	<0.001	

Reviewer: 3

Reviewer Name: Justin Manjourides

Institution and Country: Northeastern University

Please state any competing interests or state 'None declared': None declared

We would like to thank Prof. Manjourides for the comments, which we have addressed below.

Please leave your comments for the authors below

Associations between metal constituents of ambient particulate matter and mortality in England; a small area study

The authors aim to assess long-term association between components of PM10 and PM2.5 and mortality and lung cancer incidence. Land use regression models estimate exposures, while health outcomes are derived from the Office of National Statistics.

Overall, I found this paper of interest and with sound methods. I do think this work could benefit from more clarity around the following topics:

1. A slightly more expanded explanation of the exposure assignment. I'm gathering from Page 3 line 44 that participants were matched to the exposure estimate for their postal code. This could be made more explicit.

We have made this explicit in the exposure paragraph.

2. Your definition of "long term exposure" was never clearly stated.

We have added this definition right at the beginning of the introduction paragraph.

3. I'm not an exposure scientist, so I am curious as to why Zinc was included in the PM10 analysis, but not the PM25 analysis.

The reason is based on the fact that the PM2.5 and PM10 Zinc model were highly correlated. PM10 Zn model (R2 = 0.80, R2 Leave One out Cross Validation (LOOCV) = 0.77) performed better than the PM2.5 Zn model (R2 = 0.70, R2 LOOCV = 0.63), so we chose for PM10 in our analysis.

4. The model on Page 4 indicates that all PM10 constituents were included in the same model and all Pm25 constituents were included in the same model. However, the correlations between these constituents, within each PM10 and PM25, are extremely high. While this is certainly a recognized issue in air pollution research, it would be good to see perhaps single-constituent models as a comparison or sensitivity analysis.

Thank you for picking this up. We made a mistake in the formula and we re-write it correctly,

$$\log(\text{RR}_i) = \mu + \sum_{j=1}^{p-1} (\alpha_j \text{Confound}_{ij}) + \beta_1 \text{PM}_{ik} + U_i.$$

also adding in the text that μ is the model intercept. The model was fitted for each metal constituents separately, as they are highly correlated.

5. The μ in the model is not defined.

We added that it is the intercept.

6. Be more explicit that the main results presented control for the listed confounders, also detailing these control variables in the table caption.

We added in the text: The individual effect of each elemental constituents of particulate matter evaluated with the Poisson regression adjusted for confounders is displayed in Table 2 and Table S1 in Supplementary Material.

In Table 2 caption we had: Individual effects of metals, estimated with Poisson regression, on cardiovascular mortality, respiratory mortality and lung cancer incidence adjusted for tobacco weekly expenditure, IMD (index of multiple deprivation) and percentage of Asian and White population. Mean and lower and upper bounds of the credible intervals of the inter-decile relative risk (RR).

7. Throughout this manuscript I would suggest emphasizing that these are all estimates of the linear association between the log(RR) and AP exposures, as there could very well be a different non-linear association caused by threshold effects or effect modification.

We agree with the reviewer comment and we have included this in the methods section and when we reported the results.

8. Summary data in Table 1 display 10th and 90th percentile for each metal and confounder. While this is informative for the metals, to get a sense of the exposures, this is less informative for the confounders. The important comparison is what the values of the outcomes and confounders are for those individuals living in areas that have the 10th percentile of exposure vs those that individuals living in the areas that have the 90th percentile of exposure. This will help tell the story of how those high exposure areas differ from those low exposure areas on key demographic features and outcome rates. I would suggest a slight reorganization of Table 1 where the columns represent the 10th and 90th percentiles of either PM10 and PM25 or, because the correlations are so high, perhaps selecting 1 constituent of PM10 and one of PM25 as an exemplar and stratify your sample demographics and outcomes on that variable.

We have re-organized table 1, as requested.

Minor:

1. Page 4, Line 35: "... for elemental constituents of in PM10 and PM25."

We have amended this.

References

Bocchi, C.; Bazzini, C.; Fontana, F.; Pinto, G.; Martino, A.; Cassoni, F. Characterization of urban aerosol: Seasonal variation of genotoxicity of the water-soluble portion of PM2.5 and PM1. *Mutation Research/Genetic Toxicology and Environmental Mutagenesis* 2019;841:23-30

de Hoogh, K.; Wang, M.; Adam, M.; Badaloni, C.; Beelen, R.; Birk, M.; Cesaroni, G.; Cirach, M.; Declercq, C.; Dédélé, A.; Dons, E.; de Nazelle, A.; Eeftens, M.; Eriksen, K.; Eriksson, C.; Fischer, P.; Gražulevičienė, R.; Gryparis, A.; Hoffmann, B.; Jerrett, M.; Katsouyanni, K.; Iakovides, M.; Lanki, T.; Lindley, S.; Madsen, C.; Mölter, A.; Mosler, G.; Nádor, G.; Nieuwenhuijsen, M.; Pershagen, G.; Peters, A.; Phuleria, H.; Probst-Hensch, N.; Raaschou-Nielsen, O.; Quass, U.; Ranzi, A.; Stephanou, E.; Sugiri, D.; Schwarze, P.; Tsai, M.-Y.; Yli-Tuomi, T.; Varró, M.J.; Vienneau, D.; Weinmayr, G.; Brunekreef, B.; Hoek, G. Development of Land Use Regression Models for Particle Composition in Twenty Study Areas in Europe. *Environmental Science & Technology* 2013;47:5778-5786

Raaschou-Nielsen, O.; Beelen, R.; Wang, M.; Hoek, G.; Andersen, Z.J.; Hoffmann, B.; Stafoggia, M.; Samoli, E.; Weinmayr, G.; Dimakopoulou, K.; Nieuwenhuijsen, M.; Xun, W.W.; Fischer, P.; Eriksen, K.T.; Sørensen, M.; Tjønneland, A.; Ricceri, F.; de Hoogh, K.; Key, T.; Eeftens, M.; Peeters, P.H.; Bueno-de-Mesquita, H.B.; Meliefste, K.; Oftedal, B.; Schwarze, P.E.; Nafstad, P.; Galassi, C.; Migliore, E.; Ranzi, A.; Cesaroni, G.; Badaloni, C.; Forastiere, F.; Penell, J.; De Faire, U.; Korek, M.; Pedersen, N.; Östenson, C.G.; Pershagen, G.; Fratiglioni, L.; Concin, H.; Nagel, G.; Jaensch, A.; Ineichen, A.; Naccarati, A.; Katsoulis, M.; Trichpoulou, A.; Keuken, M.; Jedynska, A.; Kooter, I.M.; Kukkonen, J.; Brunekreef, B.; Sokhi, R.S.; Katsouyanni, K.; Vineis, P. Particulate matter air pollution components and risk for lung cancer. *Environment International* 2016;87:66-73

VERSION 2 – REVIEW

REVIEWER	R. Bruce Urch, Adjunct Professor Division of Occupational & Environmental Health, Dalla Lana School of Public Health, University of Toronto, Toronto, Ontario, Canada
REVIEW RETURNED	01-Aug-2019

GENERAL COMMENTS	The comments raised in the initial review were addressed satisfactorily with the exception of: Introduction. Pg 3, lines 10-12. “In the Transport related Air Pollution and Health impacts – Integrated Methodologies for Assessing Particulate Matter (TRANSPHORM) study, copper zinc and iron content of particulate matter ...” Specify what size fraction of PM (PM10, PM2.5). The same for the Wang et al study (ref 9). Revised Table 1: The metal concentration summaries in the original Table 1 was informative. I would include this in the supplement. The revised Table 1 is confusing, as there is no description of it. It should be summarized in the text, so that the reader has a better understanding of what you are trying to demonstrate (e.g., increased mortality for 90th vs 10th percentile). Also, I am not sure why PM10 Cu was selected a representative example (per reviewer 3, comment #8). The only significant association for PM10 Cu was a protective effect with respiratory mortality. In the abstract you list significant adverse associations between cardiovascular mortality and PM2.5 Cu as well as respiratory mortality and PM10 Zn. Either one of these or both would be a better choice. Report all values to 2 decimals. e.g., 0.6 should be 0.60. Other new comments/edits: Introduction. First paragraph. “Chronic exposure to toxic substances in fine particulate matter (PM) with aerodynamic diameter less than 10µm (PM10)¹⁻³ and 2.5µm (PM2.5)⁴ is associated with increased mortality levels from cardiovascular disease^{1 5}”. Fine PM is < 2.5 µm. Coarse PM is < 10 µm. Remove “fine”. Exposure data. PM was collected from 2010-2011 and elemental composition determined from 20 sites over two 3-week periods to develop LUR models. Please add a statement that LUR models were used to predict PM10 and PM2.5 metal exposures for the period 2008-2011, as detailed in the Discussion. Also, you should include this as a limitation of the study under “Strengths and limitations of this study”. Table 3. Modify the title to: Pearson inter-correlations (r) between particulate matter (PM) metals. Please list the N for this Table. Table S1. Modify title to “Poisson regression confounder effects from the two models (i) using metals from PM10 and (ii) metals from PM2.5 for all the health outcomes. Mean, lower and upper bound of the 95% credible interval (CI) of the inter-decile relative risk (RR).” Added (CI). Define IMD and VIF. Table S2. Modify title to” Pearson correlation (r) between the adjusted annual mean (24-hr) concentrations of PM-metals and the adjusted annual mean PM concentrations (PM2.5 and PM10).” Added (r) and 24-hr. If not 24-hr PM measures, specify sample period. Please list the N for this Table. What does “***” beside each correlation coefficient mean?- if significant, see below.
--

	Since all correlations were significant ($p < 0.001$), I would remove all p-values (and **) and add a statement below that all correlations were significant ($p < 0.001$). Pearson correlation can be highly influenced by extreme values. Did you plot the values to visualize the distribution? The non-parametric Spearman correlation is less influenced by extreme values and may be more appropriate if the values are not normally distributed. Discussion. “Wolf et al 20 found elevated but non-significant associations with copper, zinc and iron constituents of particulates with incident coronary events in 11 cohorts (5,157 events), while Wang et al. 9 did not find long-term associations with cardiovascular mortality (9545 deaths) in 19 European cohorts where exposure results from a single year were applied over 2-20 years follow-up, in some cases retrospectively.” Please specify the size fractions of PM.
--	---

VERSION 2 – AUTHOR RESPONSE

Response to the Comments from reviewers:

We thank Professor Urch for these useful comments and suggestions.

The comments raised in the initial review were addressed satisfactorily with the exception of: Introduction. Pg 3, lines 10-12. “In the Transport related Air Pollution and Health impacts – Integrated Methodologies for Assessing Particulate Matter (TRANSPHORM) study, copper zinc and iron content of particulate matter ...” Specify what size fraction of PM (PM₁₀, PM_{2.5}). The same for the Wang et al study (ref 9).

We have added these.

Revised Table 1:

The metal concentration summaries in the original Table 1 was informative. I would include this in the supplement.

We have added the original table 1 in supplementary material as S1.

The revised Table 1 is confusing, as there is no description of it. It should be summarized in the text, so that the reader has a better understanding of what you are trying to demonstrate (e.g., increased mortality for 90th vs 10th percentile). Also, I am not sure why PM₁₀ Cu was selected a representative example (per reviewer 3, comment #8). The only significant association for PM₁₀ Cu was a protective effect with respiratory mortality. In the abstract you list significant adverse associations between cardiovascular mortality and PM_{2.5} Cu as well as respiratory mortality and PM₁₀ Zn. Either one of these or both would be a better choice. Report all values to 2 decimals. e.g., 0.6 should be 0.60.

We have re-computed this table 1 by PM_{2.5} Cu, and reported to 2 decimal places. We have changed the example percentiles for PM_{2.5} Cu as suggested and agree that this is a more useful example.

Other new comments/edits:

Introduction. First paragraph. "Chronic exposure to toxic substances in fine particulate matter (PM) with aerodynamic diameter less than 10µm (PM10)¹⁻³ and 2.5µm (PM2.5)⁴ is associated with increased mortality levels from cardiovascular disease^{1 5}".

Fine PM is < 2.5 µm. Coarse PM is < 10 µm. Remove "fine".

We have removed it.

Exposure data. PM was collected from 2010-2011 and elemental composition determined from 20 sites over two 3-week periods to develop LUR models. Please add a statement that LUR models were used to predict PM10 and PM2.5 metal exposures for the period 2008-2011, as detailed in the Discussion. Also, you should include this as a limitation of the study under "Strengths and limitations of this study".

We have amended this in the Exposure paragraph and in the discussion.

Table 3. Modify the title to: Pearson inter-correlations (r) between particulate matter (PM) metals. Please list the N for this Table.

We have amended this and added the N =1533.

Table S1. Modify title to "Poisson regression confounder effects from the two models (i) using metals from PM10 and (ii) metals from PM2.5 for all the health outcomes. Mean, lower and upper bound of the 95% credible interval (CI) of the inter-decile relative risk (RR)."

Added (CI). Define IMD and VIF.

We have added (CI) and a footnote for the acronyms.

Table S2. Modify title to "Pearson correlation (r) between the adjusted annual mean (24-hr) concentrations of PM-metals and the adjusted annual mean PM concentrations (PM2.5 and PM10)." Added (r) and 24-hr. If not 24-hr PM measures, specify sample period.

This is now Table S3. We have changed the title to: 'Pearson Correlation(r) between the adjusted annual mean concentrations of PM-metals and the adjusted annual mean PM concentrations (PM2.5 and PM10).'

Please list the N for this Table.

We have amend this and added the N =1533.

What does "***" beside each correlation coefficient mean?- if significant, see below.

Since all correlations were significant (p< 0.001), I would remove all p-values (and **) and add a statement below that all correlations were significant (p< 0.001).

We removed the p-values and added a footnote.

Pearson correlation can be highly influenced by extreme values. Did you plot the values to visualize the distribution? The non-parametric Spearman correlation is less influenced by extreme values and may be more appropriate if the values are not normally distributed.

We did check normality assumption. Data are normally distributed. We used a linear regression model with land use covariates (land use regression) and the linear regression assumption is based on data being normally distributed.

Discussion. “Wolf et al 20 found elevated but non-significant associations with copper, zinc and iron constituents of particulates with incident coronary events in 11 cohorts (5,157 events), while Wang et al. 9 did not find long-term associations with cardiovascular mortality (9545 deaths) in 19 European cohorts where exposure results from a single year were applied over 2-20 years follow-up, in some cases retrospectively.” Please specify the size fractions of PM.

We have added these.

VERSION 3 – REVIEW

REVIEWER	R. Bruce Urch, Adjunct Professor Division of Occupational and Environmental Health, Dalla Lana School of Public Health, University of Toronto, Toronto, Ontario, Canada
REVIEW RETURNED	27-Sep-2019

GENERAL COMMENTS	Previous comments & suggestions were addressed, however, there are still some issues with Table 1. Table 1: “Descriptive statistics of health outcomes, modelled particulate metal concentrations, deprivation score, and eth-nicity covariates for the 1533 wards in the study area in 2008-11, all stratified by PM2.5 Copper >10th, 10th-90th and >90th quantile.” “Stratified” is a statistical term. This may cause some confusion if used. You are only reporting descriptive data for <10th, 10th-90th and >90th quantiles of PM2.5 copper. You could use subdivided or partitioned or just say descriptive data are shown for <10th ... copper. Please modify in table and text. Note: >10th should be <10th. Please modify in table and text. I assume that you are also reporting descriptive data for <10th, 10th-90th and >90th quantiles of PM2.5 copper for the area-level confounders. However, the modelled metal concentrations show descriptive data for <10th, 10th-90th and >90th quantiles, not quantiles of PM2.5 copper. I would remove the modelled metal concentrations from Table 1. It is just adding confusion. Furthermore, you have descriptive data for the metals in Table S1. In Table 1, the R2 for Fe PM10 is 0.77 but it is 0.95 in Table S1 and the R2 for Zn PM10 is 0.95 in Table 1 and 0.77 in Table S1. Please check your data. Also the data for Zn PM10 are much higher in Table S1 vs Table 1. Please check your data. Also, three minor edits. Table 1. The footnote “1” for CU PM2.5 is “1 Cu PM10 Metals in ng/m3 LOOCV R2=0.79.” Change PM10 to PM2.5. Table 3 title, “Pearson inter-correlation(r) between the particle metals (PM) metals (n=1533).” Title should read Pearson inter-correlation(r) between the particulate matter (PM) metals (n=1533). Change “particle metals” to “particulate matter”. Table S3. Be consistent with the format in Table 3. i.e. Report correlations to 2 decimals and include 0 before decimal, e.g. 0.86. The “***” is distracting beside every correlation and not needed as
---

	you report this in the footnote. Change footnote to “All correlations significant at $p < 0.001$.”
--	---

VERSION 3 – AUTHOR RESPONSE

Reviewer: 1

Reviewer Name: R. Bruce Urch, Adjunct Professor

Institution and Country: Division of Occupational and Environmental Health, Dalla Lana School of Public Health, University of Toronto, Toronto, Ontario, Canada

Please state any competing interests or state 'None declared': None declared

We thank Professor Urch for these useful comments and suggestions.

Previous comments & suggestions were addressed, however, there are still some issues with Table 1.

Table 1: “Descriptive statistics of health outcomes, modelled particulate metal concentrations, deprivation score, and ethnicity covariates for the 1533 wards in the study area in 2008-11, all stratified by PM2.5 Copper >10th, 10th-90th and >90th quantile.”

“Stratified” is a statistical term. This may cause some confusion if used. You are only reporting descriptive data for <10th, 10th-90th and >90th quantiles of PM2.5 copper. You could use subdivided or partitioned or just say descriptive data are shown for <10th ... copper. Please modify in table and text.

Note: >10th should be <10th. Please modify in table and text.

We changed as suggested to the following:

Descriptive statistics of health outcomes, modelled particulate metal concentrations, deprivation score, and ethnicity covariates for the 1533 wards in the study area in 2008-11, subdivided by PM2.5 Copper <10th, 10th-90th and >90th quantile.

I assume that you are also reporting descriptive data for <10th, 10th-90th and >90th quantiles of PM2.5 copper for the area-level confounders.

However, the modelled metal concentrations show descriptive data for <10th, 10th-90th and >90th quantiles, not quantiles of PM2.5 copper. I would remove the modelled metal concentrations from Table 1. It is just adding confusion. Furthermore, you have descriptive data for the metals in Table S1.

We have removed the modelled data from table 1, as suggested.

In Table 1, the R2 for Fe PM10 is 0.77 but it is 0.95 in Table S1 and the R2 for Zn PM10 is 0.95 in Table 1 and 0.77 in Table S1. Please check your data. Also the data for Zn PM10 are much higher in Table S1 vs Table 1. Please check your data.

We have checked the R2 in both tables and the one from table S1 are correct. The typo was in Table 1, but the metals have now been removed.

Also, three minor edits.

Table 1. The footnote “1” for CU PM2.5 is “1 Cu PM10 Metals in ng/m3 LOOCV R2=0.79.” Change PM10 to PM2.5.

We have corrected this.

Table 3 title, “Pearson inter-correlation(r) between the particle metals (PM) metals (n=1533).” Title should read Pearson inter-correlation(r) between the particulate matter (PM) metals (n=1533). Change “particle metals” to “particulate matter”.

We have changed this.

Table S3. Be consistent with the format in Table 3. i.e. Report correlations to 2 decimals and include 0 before decimal, e.g. 0.86. The “***” is distracting beside every correlation and not needed as you report this in the footnote. Change footnote to “All correlations significant at $p < 0.001$.” We have amended it.

VERSION 4 – REVIEW

REVIEWER	R. Bruce Urch, Adjunct Professor Division of Occupational & Environmental Health, Dalla Lana School of Public Health, University of Toronto, Toronto, Ontario, Canada
REVIEW RETURNED	24-Oct-2019

GENERAL COMMENTS	The latest changes made were satisfactory. The manuscript is acceptable once the following two edits are made. 1) Table 1. As you have removed the modeled metal data from Table 1, you will need to modify the Table title and remove “modeled particulate metal concentrations”. 2) Results. 2nd sentence. “We have reported summary descriptive statistics for Standard Mortality/Incidence Rates (SMR/SIRs), metal constituents of PM and confounders, stratifying the wards between the 10th percentile of exposure 90th percentile of PM2.5 Copper.” Table 1 shows descriptive statistics subdivided by PM2.5 copper. Table S1 does not, it shows descriptive statistics for the 10th and 90th centile. I would modify the sentence by removing the end part “stratifying the wards between the 10th percentile of exposure 90th percentile of PM2.5 Copper”. i.e., “We have reported summary descriptive statistics for Standard Mortality/Incidence Rates (SMR/SIRs), metal constituents of PM and confounders.” In the sentence that follows you report on the 90th vs 10th percentiles of PM2.5 copper, just remove metal constituents of PM. i.e. SMRs/SIRs, area-level deprivation, non-white ethnicity and tobacco sales (smoking proxy) were all higher in wards in the 90th vs. 10th percentile PM2.5 copper.
--